# The Role of Seawater and Saline Solutions in Treatment of Upper Respiratory Conditions

**DOI:** 10.3390/md20050330

**Published:** 2022-05-17

**Authors:** Danijela Štanfel, Livije Kalogjera, Sergej V. Ryazantsev, Kristina Hlača, Elena Y. Radtsig, Rashidov Teimuraz, Pero Hrabač

**Affiliations:** 1Jadran-Galenski Laboratorij d.d., Svilno 20, 51000 Rijeka, Croatia; 2Department of Otorhinolaryngology, Head and Neck Surgery, Sisters of Mercy University Hospital, 10000 Zagreb, Croatia; kalogjera@sfzg.hr; 3National State Research Institute of Ear, Throat, Nose and Speech, 190013 St. Petersburg, Russia; professor.ryazantsev@mail.ru; 4Morozov Children’s City Clinical Hospital, Russian National State Pirogov Medical University, 117437 Moscow, Russia; radena@rambler.ru; 5Regional Center of Traumatology and Ortopedics, Department of Otorhinolaryngology, Karaganda State Polyclinic, 100000 Karaganda, Kazakhstan; x461yom@mail.ru; 6“Andrija Štampar” School of Public Health, School of Medicine, University of Zagreb, 10000 Zagreb, Croatia; pero.hrabac@mef.hr

**Keywords:** seawater, seawater preparation, nasal irrigation, upper respiratory track, otorhinolaryngology

## Abstract

The history of saline nasal irrigation (SNI) is indeed a long one, beginning from the ancient Ayurvedic practices and gaining a foothold in the west at the beginning of the 20th century. Today, there is a growing number of papers covering the effects of SNI, from in vitro studies to randomized clinical trials and literature overviews. Based on the recommendations of most of the European and American professional associations, seawater, alone or in combination with other preparations, has its place in the treatment of numerous conditions of the upper respiratory tract (URT), primarily chronic (rhino)sinusitis, allergic rhinitis, acute URT infections and postoperative recovery. Additionally, taking into account its multiple mechanisms of action and mounting evidence from recent studies, locally applied seawater preparations may have an important role in the prevention of viral and bacterial infections of the URT. In this review we discuss results published in the past years focusing on seawater preparations and their use in clinical and everyday conditions, since such products provide the benefits of additional ions vs. saline, have an excellent safety profile and are recommended by most professional associations in the field of otorhinolaryngology.

## 1. Introduction

The use of water for prophylactic or therapeutic purposes, mostly in the respiratory system, has been known since ancient times. In Yogic practices, different nasal cleansing techniques are used as part of a wider range of body-cleansing procedures. Vedic texts describe several techniques called “neti” [1,2], with “jala neti” [3,4] corresponding to today’s concept of nasal cavity irrigation. In the neti techniques, copperware is used for irrigation (to prevent contamination of the solution), the solution is heated to body temperature and an exact salt concentration in the preparation of the solution is specified. This salt content, and consequently, the osmolality of the solution, remains one of the most important parameters in nasal irrigation to the present day.

The osmolality of the commercial compositions of NaCl solution ranges from the physiological level (0.9%) to the hypertonic level with an osmolality of 3% [2]. Solutions with higher osmolality tend to induce side effects, such as nasal burning, blockage and dripping [5]. The osmolality of the solution results not only from NaCl content but also from the other ions contained therein. Besides having effect on osmolality, the ions also show a number of effects on the biology and function of cells and tissues. This is especially important because many commercially available formulations, primarily those based on seawater, contain a number of ions other than Na^+^ and Cl^−^, and differ significantly from the galenic saline. In this review, the following parameters of these solutions will be discussed:Composition of solution in context of differences between saline and solutions based on seawater.Mechanism of action in nasal cavity and elsewhere.Safety and efficacy of use in different indications.

## 2. Composition of Saline/Seawater Preparations

Unlike saline, which consists of NaCl dissolved in distilled water, in seawater there are four categories of constituents or solutes: major constituents, minor constituents, trace elements and gases. Average salinity of undiluted seawater is approximately 3.5%, or 35 ppt (parts per thousand). Ninety-nine percent of seawater salinity is due to six major constituents: Cl^−^, Na^+^, SO_4_^2−^, Mg^2+^, Ca^2+^ and K^+^. Salinity is relatively uniform, with ranges of variation of 33–37 ppt in open ocean water, 37–38 ppt in smaller bodies of seawater, such as the Adriatic Sea, and as much as 240 ppt in the Dead Sea. This is why the source of the water in seawater products is such an important factor.

One of the fundamental laws in oceanography, the Forchhammer principle, or the principle of constant proportions, states that the relative proportions of the major constituents of seawater are constant, regardless of different salinities in different sea-water samples [6]. Cl^−^ accounts for 55% of the ions, followed by Na^+^ (30.6%), SO_4_^2−^ (7.7%) and Mg^2+^ (4%). Major constituents are also considered to be conservative, i.e., chemically non-reactive and thus stable in oceans and seas over long periods of time. The oceans have a major impact on climate regulation and climate changes, the life cycle of nutrients in nature, energy flow and the biodiversity of its inhabitants. The chemical and physical properties of oceans are changing, with global warming and acidification due to increased carbon absorption. Warmer air can absorb more water than colder air, so as the climate heats up, more water can evaporate into the air, as a result of which salinity increases [7,8] but as mentioned before, different salinities do not affect the relative portions of the major constituents. Besides the major constituents measured in ppt, seawater also contains a number of minor constituents, measured in ppm—parts per million, and trace elements, measured in ppb—parts per billion. However, the principles that apply to the major elements do not apply to the minor and trace elements. This means that many of these elements are biologically or chemically reactive, and that their concentration can be dependent on biological activity and other factors, exhibiting significant local differences. The major constituents of seawater with salinity of 35 ppt at the temperature of 25 °C are shown in Table 1.

From Table 1, it is evident that the cations (sodium, potassium, calcium and magnesium) determined in seawater from the Adriatic Sea (Kvarner bay) by the ion-chromatography method are higher at higher levels than cations obtained by various authors mentioned in Table 1.

Another parameter of primary importance for seawater products is osmolality. Seawater with concentrations of approximately 26% to 27% becomes hypertonic (considering the plasma osmolality reference range of 285–295 mOsm/kg [14] and can exert a range of effects associated with hypertonic solutions.

## 3. Mechanism of Action

The mechanism of action of nasal irrigation solutions is based on two principles: physical and biological/physiological. The first principle is based on the physical (mechanical) effect of cleansing the nasal mucosa of the accumulated secretion and pathogens. The second principle depends on the effects of the ions on the physiology of the mucosal cells. In the Figure 1 we propose the chain of events following mucosal application of seawater preparations which results in a range of beneficiary effects.

The mechanism displayed in Figure 1 centers on the transport of water through the mucosal epithelial membrane, provoked by the local application of hypertonic solution. The nasal mucosa is hydrated and moisturized by both the local application of solution and the influx of water through the membrane. This leads to increased mucociliary clearance (MCC) [15,16]. Additionally, liquid transport through the membrane results in the accumulation of liquid in the nasal lumen, which concept was also proven in other organs and tissues, both in vitro [17,18] and in vivo [19,20]. Reduction in swelling (oedema) is seen in submucosal tissue, while the immediate effect of excess liquid in the nasal lumen is mechanical cleaning of mucus, crusts and debris. Immediately following this, the state of the mucus changes from gel to sol [21]. The transition of mucus from gel to sol state greatly reduces the amount of energy needed by cilia to transport such mucus [22], significantly improving the efficacy of the mucociliary transport. Additional ionic constituents of seawater show other effects, such as increased cell viability and inflammation reduction (Figure 1 and Table 2).

Mucociliary transit time (MTT; the time needed for a compound to be transported a certain distance within the respiratory system) is used to assess the efficacy of mucociliary clearance. Compared to healthy volunteers with mean MTT of 12.01 ± 3.0 min, this time is significantly prolonged in subjects with a history of allergic rhinitis (15.5 ± 3.5 min) and in heavy smokers (16.5 ± 5.0 min) [23]. Similarly, it has been shown that patients with a wide variety of diseases, ranging from septum deviations [24] to chronic sinusitis [25], have prolonged MTT, and that the restoration of mucociliary clearance is of significant importance in treating the disease [26]. Therefore, the efficacy of mucociliary transport might be one of the key mechanisms in the positive effect of nasal irrigation solutions on the nasal tissue [27,28]

On the most basic level, MTT depends on the ciliary beat frequency (CBF). Wabnitz et al. used nasal sprays with 0.9% and 3.0% sodium chloride on eight healthy volunteers with a mean baseline CBF of 9.6 Hz. While isotonic saline reduced the CBF firstly to 9.1 Hz (after 5 min) and then to 8.8 Hz (after one hour), use of 3.0% saline increased the CBF to 10.1 Hz before it returned to near-baseline levels (9.2 Hz) at 60 min [29]. Similar results were seen when monitoring another parameter, saccharine clearance time, which decreased from a median of 11.17 min to a median of 6.83 after application of isotonic saline, and one of 7.14 min after application of hypertonic saline [30]. These results, which show a beneficiary effect with hypertonic saline, but a much smaller or completely absent effect with isotonic saline, are confirmed by other authors [31,32,33,34]. The same effects of hypertonic saline were shown for mucociliary clearance in asthmatic patients [35], subjects with cystic fibrosis [36,37,38], children with bronchiolitis [39] and healthy subjects [19]. On the molecular level, this effect of hypertonic saline seems to be based on the upregulation by of CLC-3, a chloride channel that accounts for the transport of chloride ions in numerous tissues and plays a fundamental role in transepithelial salt and water movement [40].

Besides the abovementioned mechanism involved in the physical and osmotic effects of the solution, different ions in seawater have a number of additional effects. These effects are displayed in Table 2.

The abovementioned findings show that, besides the immediate positive effect of the mechanical cleaning of the mucosal surface, there is an additional and potentially more important positive effect exerted through the facilitation of the physiological function of mucociliary transport achieved by a saline solution of adequate osmolality. Additionally, other ions contained in the solution show a wide range of beneficial physiological effects on a cellular level.

## 4. Aspects of Saline/Seawater in Human Use

Table 3 shows the main safety and efficacy conclusions from clinical trials and in-vitro studies performed over more than 20 years. We searched the MEDLINE, Scopus, Web of Science and Cochrane databases to identify studies of interest. The aim was to identify as much (especially clinical) studies as possible. To achieve this, we used a broad search strategy, including only the basic keywords of “seawater” and “saline”. For example, a MeSH search syntax was “Seawater”[Mesh] OR “Saline Solution”[Mesh] OR “Saline Solution, Hypertonic”[Mesh]. Because MeSH indexing takes some time, an additional PubMed search with the same keywords was performed for the studies published over the last three years. Additional studies were identified through Scopus, and in particular, by following “Times Cited” links for the Web of Science results. After examining all the identified studies, we focused on those that, in our opinion, contributed most to the understanding of the safety and efficacy aspects of nasal irrigation use in human medicine. Studies with both seawater and saline solutions in a wide range of osmolalities and compositions were covered. The safety and efficacy of these preparations will be shortly discussed here.

### 4.1. Safety

The safety of preparations based on both saline and seawater has been proven in numerous studies, with subjects ranging from healthy individuals to infants and pregnant women. More than 60 such studies are listed in Table 3, covering the period of last 23 years. General side effects are rare, while serious side effects are virtually non-existent. Moreover, one must take into account the fact that in most of the studies, subjects had at least one additional condition or diagnosis, such as allergic rhinitis, rhinosinusitis, postoperative status, asthma, bronchiolitis, etc. Most of these conditions require additional therapy, which in itself could be the reason for the side effect(s) ascribed to nasal irrigation treatment. In most of the abovementioned studies, adverse events were either not mentioned in the text of the papers, or were not reported by study participants. In cases where adverse events were mentioned, most pertained to a burning feeling in the nose and throat. Some studies report that the incidence of this adverse event is rather high; for instance, a mild burning sensation was reported by a majority (57% [35]) of subjects in a study by Kumar et al. In the same study, moderate burning, as opposed to mild, was much less pronounced, with only 19% subjects reporting this side effect. Furthermore, the intensity of the burning sensation seemed to be correlated to osmolality of the preparation, with hypertonic preparations causing more adverse events. Other studies report similar rates of burning among their participants. Shoseyov et al. [42] describe burning sensations in four (of a total of 34) paediatric subjects with chronic sinusitis, with three taking hypertonic saline and one taking an isotonic preparation (note a similar rate of adverse events between hypertonic and isotonic groups described in the previous study. However, there are studies where this rate is inverse [28]. Other studies mentioning burning sensations as a side effect of nasal irrigation therapy either fall within the incidence boundaries described above [58,72] or discuss burning as a side effect not affecting subjects’ participation in the study or the study’s outcome [33,57,62,66].

Other adverse events were rare, and included nasal drainage [57,60], epistaxis [41,62,84], bitter taste in mouth [62], pain [92] and nose dryness [50].

### 4.2. Efficacy

The efficacy of nasal irrigation solutions has been proven in numerous clinical trials and studies, most of which are listed in Table 3 and Table 4. Efficacy has been proven in a variety of populations, from pregnant women and children to adults with a wide range of pathological conditions. Given that the attached list of publications speaks for itself, we concentrate here on presenting the essential facts about a few of the most important indications.

Regarding the exact posology, the question remains an open one. There are numerous factors affecting the dosage to be administered. Detailed data are laid out in the tables below, with the basic elements to be considered being age (paediatric vs. adult), indication (allergy, sinusitis, postoperative indications or usage in healthy individuals), product concentration (isotonic or hypertonic, up to approximately 3.5%) and methodology of application (drops, spray, nebulizer or irrigation). Therefore, the dosage may range from a few drops of hypertonic solution in children with URTI to extensive lavage with isotonic solution in adult subjects with a variety of indications.

#### 4.2.1. Chronic Sinusitis

By definition, chronic rhinosinusitis (CRS) is an inflammation of the paranasal sinuses seen in several percent of both the paediatric and adult populations [104]. The diagnosis is based on the presence of at least two of four cardinal symptoms for at least 12 weeks, and is confirmed by physical examination and (if necessary) additional radiological methods [105]. Intranasal spray administration of corticosteroids is known to significantly improve symptoms, and a similar consensus exists for nasal saline irrigation. The use of oral antibiotics may be indicated in cases of acute exacerbations of the disease, although this was not corroborated in the recent Cochrane review on this topic [99]. Similar results have been described in children by a group of Russian authors [106].

Papers listed in Table 3 and Table 4 strongly confirm these findings. In a paediatric population, Pham et al. [78] showed that 6-week treatment is well tolerated in children and is useful both as a first-line treatment for CRS and as an effective measure reducing the need for surgery. Regarding tonicity, in another paediatric study, hypertonic solution was shown to be comparable to the isotonic variety in terms of safety, although the number of adverse events was higher in the hypertonic group [42].

Evidence of both safety and efficacy is, expectedly, more abundant in adult populations. Subjects treated with nasal saline used fewer antibiotics compared to the control group [47] and hypertonic solution was reported as superior to isotonic solution [67,73,88]. Other hypertonic saline preparations, such as Dead Sea salt, have also been proven as safe and effective in this indication [56]. While various application methods are used (mostly spray vs. low/large volume irrigation [46,60], the safety profile remains highly favourable across the various studies.

#### 4.2.2. Allergic Rhinitis

Allergic rhinitis is an extremely common condition that is also commonly overlooked in the diagnostic process, resulting in significant public health effects. Additionally, although it is not a severe illness, allergic rhinitis can significantly complicate the symptoms, diagnosis and clinical course of other diseases [107].

Nasal irrigation preparations have been shown to be effective [16] and safe [70] as both long-term [83] and short term [74] treatments, and to reduce the need for other commonly used treatment options, such as antihistamines, in children [48,53] and pregnant women [108]. The same was proven for the use of nasal steroids [65,77] and systemic drugs [49].

#### 4.2.3. Other Indications

Besides the two major indications listed above, there are numerous studies in other indications, as well as in vitro studies [54] and studies performed on healthy participants, with the latter serving primarily as the proof of concept for the safety and efficacy of nasal irrigation treatments.

Different methods of saline penetration were tested using the Technetium-99 labelled solution, with douching being the method with best penetration in the maxillary sinus [51]. Positive effects of nasal irrigation were proven in healthy army conscripts [50], adult subjects [18,29,45] and otherwise healthy subjects exposed to wood dust [41,43].

Regarding other indications, positive effects were described in paediatric patients with viral bronchiolitis [59], bronchiolitis in the intensive care unit [86], acute sinusitis [80], acute upper respiratory tract infections [82,85], chronic tonsilitis [61] and cold and influenza [62]. Moreover, daily nasal irrigation in the paediatric stage (especially in children who cannot blow their noses) is a practice that should be encouraged as a good habit, even without underlying pathologic conditions. Similar studies exist regarding adult subjects [55,89], including pregnant women [84]. Studies on the postsurgical beneficiary effects of saline solutions [28,75], retrospective studies [109] and those based on questionnaires and surveys [66,76] seem to confirm all of the above-mentioned effects.

#### 4.2.4. The Place of Saline/Seawater Preparations in the COVID-19 Pandemic

Finally, although it is too early to speculate on whether the use of nasal irrigation solutions has a place in preventing or reducing the symptoms of viral infections [110], a recent publication on people infected with coronavirus [89] suggests that this could be an interesting area of research in the near future. Additionally, there is a growing number of papers on this topic, suggesting the potential positive effects of saline irrigations during the pandemic, both as preventive [111,112,113] and a treatment option [89]. In a recent paper, a multidisciplinary group of Belgian authors [114] proposed a detailed hypothesized mechanism of action of saline in coronavirus infections. The mechanism is similar to that proposed in the present article, including, among other aspects, wetting effects on the local tissue, mucus gelling, and the effects of the increased NaCl concentration on mucosa. Due to the effects described earlier in this paper and elsewhere [115], if used early and as an add-on therapy, locally applied nasal irrigation solutions may represent an interesting and promising remedy for all viral infections, including SARS-CoV-2 [116].

## 5. Conclusions

Nasal irrigation solutions show numerous positive effects in clinical use in the upper respiratory tract. These are mainly mechanical (cleaning of the mucosa) and related to osmolality (oedema reduction and moisturizing of the epithelium). In our paper, we presented a comprehensive body of evidence regarding the beneficiary effects of nasal irrigation solutions in general as well as for a wide variety of clinical indications, such as infectious diseases of the upper respiratory tract, allergic rhinitis, postoperative care, etc. All information mentioned above, especially the data in Table 2 [3,20], clearly favours seawater preparations over saline. However, a definitive recommendation can be given only after the careful evaluation of EBM levels for each of the papers discussed. Due to its chemical constituents, such as magnesium, calcium, potassium, bicarbonate and other ions, seawater shows a range of additional chemical effects, from promoting cell repair and reducing inflammation to reducing viscosity of the mucus and increasing ciliary beat frequency. Numerous studies in URT patients, pregnant women, children and elderly individuals show exceptionally good safety profiles for seawater preparations [82,98,108,117]. Side effects are rare, and consist mostly of burning feelings and nasal drainage, with serious adverse events practically non-existent.

To the best of our knowledge, a scientifically proven consensus on the exact mechanism of action of seawater in the human upper respiratory tract does not exist. Therefore, based on a comprehensive literature search, we propose a mechanism of action considering all the different aspects of seawater solution(s), from chemical composition to pH and tonicity. Further studies will be needed to confirm the present findings.

## Figures and Tables

**Figure 1 marinedrugs-20-00330-f001:**
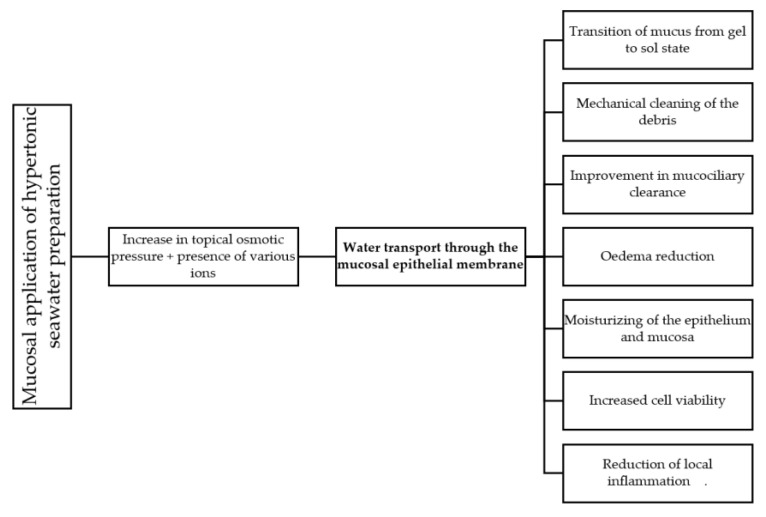
Proposed mechanism of action of hypertonic seawater preparations locally applied to mucosa of the upper respiratory tract.

**Table 1 marinedrugs-20-00330-t001:** Major constituents of seawater (mg/dm^3^).

Constituent	Dittmar (1940) * [9]	Cox (1966) * [10]	Riley (1967) * [11]	Millero (1996) * [12]	Štanfel (2006) * [13]
Cl^−^	19,805	-	-	19,805	19,763
Na^+^	11,015	11,013	11,037	11,035	12,117
SO_4_^2−^	2764	-	2776	2764	2707
Mg^2+^	1327	1327	1322	1314	1417
Ca^2+^	418	422	422	422	474
K^+^	397	408	408	408	443
Br^−^	67	-	69	69	63

* Seawater sample source. Dittmar (1940): various parts from Atlantic, Indian and Pacific Ocean during the “Challenger” expedition; Cox (1966): locations from the world’s oceans (Atlantic, Indian, Pacific). Detail locations of the samples can be seen in Cox (1996) [10]; Riley (1967): Mediterranean, Irish, Baltic Sea; Millero (1996): Baltic Sea, Baltic-North Sea, Red Sea; Štanfel (2006): Adriatic Sea Kvarner Bay.

**Table 2 marinedrugs-20-00330-t002:** Mechanism of action of other constituents in seawater [3,20].

Constituent	Action
Mg^2+^	Promotes cell repair and limits inflammation by reducing the eicosanoid metabolism both at the level of the liberation of arachidonic acid and by direct inhibition of the 5-lipoxygenase enzyme. Inhibits exocytosis from permeabilized eosinophils. Reduces apoptosis of respiratory cells.
Ca^2+^	Acetylcholine and serotonin act as messengers, increasing calcium intake in ciliated cells and thus regulating ciliary beat frequency and synchronization. Airflow promotes cell calcium intake and ciliary beat via shear-stress-induced mechanotransduction.
K^+^	Anti-inflammatory action. Promotes respiratory epithelium repair via the EGF/EGFR pathway.
HCO_3_^−^	Reduces mucous viscosity by acting as a buffer. Facilitates elimination by ciliary cells movement.

**Table 3 marinedrugs-20-00330-t003:** Overview of safety and efficacy conclusions from studies with saline and/or saltwater.

Study	Design	Subjects	Intervention	Safety Conclusions	Other Remarks
Holmstrom, 1997 [41]	Cross-sectional	45 healthy adults exposed to wood dust	Nasal lavage with Rhinomer force 2, four times a day, every workday.	One increase in allergic symptoms (with concomitant local steroid). One anterior epistaxis and one throat irritation.	At week 3, 88% subjects wanted to continue treatment, and 3 weeks after stopping treatment, 83% wished to start the treatment again.
Shoseyov, 1998 [42]	RCT	34 children with chronic sinusitis	Hypertonic (3.5%) vs. isotonic saline, 10 drops, three times daily for 4 weeks.	Three subjects in hypertonic group and one in isotonic group left study because of the burning feeling in the nose and throat.	Burning and itching was more common in hypertonic group, but only during the first 3 to 4 days. After that period, there was no difference between the groups.
Rabone, 1999 [43]	Crossover trial with 1-year follow-up	46 woodworkers exposed to wood dust	Gravity fed, home-made unbuffered isotonic saline for 2 months.	Generally safe, no notable adverse events.	The group reported significantly decreased nasal symptoms and over half of subjects continued to use nasal lavage voluntarily after 1 year.
Taccariello, 1999 [44]	RCT	40 patients with chronic rhinosinusitis	Traditional alkaline nasal douche vs. a sterile seawater spray, in addition to their regular treatment.	No adverse effects mentioned.	Alkaline nasal douche had a significant effect upon endoscopic appearances, whereas the spray did not; conversely, spray improved quality of life, whereas alkaline douche did not.
Bachmann, 2000 [45]	RCT	40 adults with paranasal sinus disease	Isotonic Ems salt solution or isotonic sodium chloride solution nasal irrigation twice daily for 7 days.	No adverse events mentioned in either group.	A slight difference between treatment with Ems salt solution and sodium chloride solution; questionable clinical relevance.
Heatley, 2001 [46]	Prospective RCT	150 adults with chronic rhinosinusitis	Nasal saline irrigation with bulb syringe or irrigation pot vs. placebo, daily for 2 weeks.	No significant adverse events; comparable efficacy in all three groups.	More than one-third of subjects reported using less concomitant medication.
Rabago, 2002 [47]	RCT	76 adults with acute or chronic rhinosinusitis	Nasal saline irrigation with 150 mL daily per nostril for 6 months vs. no treatment.	Ten side effects, of which 8 were considered as “not significant” and 2 as significant, but affected subjects were still “highly satisfied” with the treatment.	Subjects treated with nasal saline used statistically significantly less antibiotic treatment compared to control group.
Garavello, 2003 [48]	Prospective RCT	20 children with allergic rhinitis	Hypertonic saline in 10 subjects, no treatment in 10 subjects; 2.5 mL in each nostril three times daily for 6 weeks.	No patients lost to follow up and no adverse events reported.	Statistically significant decrease in use of oral antihistamines in hypertonic saline group.
Lee, 2003 [33]	RCT, crossover	28 healthy adult subjects	Hypertonic (Sinomarin, 3%) or isotonic saline. 10 sprays of both preparations (on different days) in the same nostril.	Complaints of mild prickling sensation after nasal douching with hypertonic seawater.	The effect of the hypertonic solution is probably due to changes in mucus viscoelastic properties.
Chkhartishvili, 2004 [49]	Case-control open clinical trial	30 children with allergic rhinitis, acute and chronic bacterial rhinosinusitis and 30 children in control group	“Aqua Maris” seawater solution, either irrigation or 2 drops in nasal cavity 3 times a day from 2 to 4 weeks.	Nasal drops in children up to 2 years of age were well-tolerated, with no complication. No adverse effects mentioned for the irrigation group.	In subjects with bacterial rhinosinusitis, time to relief of symptoms in Aqua Maris group was 7 ± 3.2 days vs. 10 ± 2.4 days in control group. In allergic rhinitis group, Aqua Maris reduced the use of systemic drugs in 7 of 15 patients.
Tano, 2004 [50]	Prospective trial	108 healthy army conscripts	10-week nasal spraying with physiological saline twice daily, followed by a 10-week period of follow up.	Two cases of nose dryness.	There was a mean of 0.7 episodes of upper respiratory tract infection during the spray period, compared with 1.0 episodes during the observation.
Wormald, 2004 [51]	Prospective, cross-over study	12 adult subjects	Nasal irrigation with normal saline containing Technetium 99m sulfur colloid	No adverse effects mentioned.	The nasal cavity was well irrigated using three techniques (spray, nebulization, douching). Douching was significantly more effective in penetrating the maxillary sinus and frontal recess. The sphenoid and frontal sinuses were poorly irrigated by all three techniques.
Cordray, 2005 [52]	Prospective, randomized, single-blind, placebo-controlled	15 patients with seasonal allergic rhinitis	Intranasal hypertonic dead sea saline spray, intranasal aqueous triamcinolone spray, placebo nasal saline spray for 7 days.	Two subjects withdrew for adverse events (unknown group).	Significant improvements were seen in both active-treatment groups; the corticosteroid spray was the more effective. Dead Sea saline solution can be an effective alternative in mild-to-moderate allergic rhinitis, particularly with respect to nasal and eye symptoms. It improves mucociliary clearance, while Mg cation probably exerts anti-inflammatory effects on the nasal mucosa and on the systemic immune response.
Garavello, 2005 [53]	Prospective RCT	44 children with allergic rhinitis	Hypertonic saline vs. no treatment; 3 sprays (50 μL) in each nostril three times daily for 7 weeks.	No adverse events in the treatment group.	Statistically significant decrease in use of oral antihistamines in hypertonic saline group.
Kim, 2005 [54]	In vitro study	Cell cultures of fully differentiated passage-2 normal human nasal epithelial cells	Cells in the cultures were treated with pure water and with 0.3% (hypotonic), 0.9% (isotonic) and 3% (hypertonic) saline solutions.	In vitro study.	mRNA for major airway mucins analysis and morphologic analysis suggest that pure water damaged epithelial cells, and that only isotonic saline did not affect their morphology.
Passali, 2005 [55]	RCT	200 patients with acute viral rhinosinusitis	Atomized nasal douche vs. nasal lavages with isotonic sodium chloride solution.	No adverse effects mentioned.	Atomized nasal douches significantly improved inspiratory and expiratory rhinomanometric resistance and nasal volumes and normalized mucociliary transport time to a physiological level.
Wabnitz, 2005 [29]	In-vitro study	8 healthy adult subjects	One application of four sprays of hypertonic (3.0%) saline (one nostril) and isotonic saline (another nostril) in all subjects.	No adverse effects mentioned.	Cell samples from subjects receiving saline solutions. The administration of hypertonic saline results in a significantly faster CBF 5 min (but not 60 min) after administration.
Friedman, 2006 [56]	Randomized, prospective, double-blind study	42 adults seeking treatment for chronic rhinosinusitis	Nasal irrigation using hypertonic dead sea salt solution with hypertonic saline.	No adverse effects mentioned.	Both groups had significant improvement after treatment. However, the dead sea salt patients had significantly better symptom relief and showed improved RQLQ(S) scores.
Rabago, 2006 [57]	Semi-structured, in-depth interviews in a 3-part, multimethod study	28 subjects with frequent rhinosinusitis and chronic sinonasal symptoms.	Hypertonic saline nasal irrigation.	Side effects including saline drainage, nasal burning and irritation were noted, but not identified as important enough to stop the treatment.	This is a well-tolerated, inexpensive, effective long-term therapy that patients can use at home with minimal training and follow-up.
Hauptman, 2007 [58]	RCT	80 adult patients with rhinosinusitis	1 mL of physiological or hypertonic saline to one nostril.	Increased nasal burning/irritation with hypertonic saline compared to physiological saline.	Buffered physiological saline significantly affected nasal airway patency, whereas buffered hypertonic saline had no effect on nasal patency.
Kuzik, 2007 [59]	Prospective, randomized, double-blinded, controlled, multicenter trial	96 infants with viral bronchiolitis	Repeated doses of nebulized 3% hypertonic saline or 0.9% normal saline, in addition to routine therapy.	All participants tolerated therapy without apparent adverse effects, and were eventually discharged after achieving full recovery.	Clinically relevant reduction in length of stay to 2.6 ± 1.9 days in hypertonic saline group, compared with 3.5 ± 2.9 days in the normal saline group.
Pynnonen, 2007 [60]	Prospective RCT	127 adults with chronic nasal and sinus symptoms	Irrigation with large volume and low positive pressure or spray for 8 weeks.	Forty-one subjects reported a total of 67 adverse effects. Posttreatment nasal drainage was the most common adverse effect (n = 14) in each group.	Nasal irrigations performed in large volumes and delivered with low positive pressure are more effective than saline sprays for treatment of chronic nasal and sinus symptoms in a community-based population.
Karpova, 2008 [61]	Open-label parallel-group trial	84 children with chronic tonsilitis	Experimental group with 64 subjects using Aqua Maris seawater solution, and control group using furacilin solution for 6–8 courses of crypt lavage.	No adverse effects mentioned.	Aqua Maris group showed superior results in terms of odynophagia and dysphagia severity and duration, and hyperaemia and infiltration of the palatine arches.
Slapak, 2008 [62]	Prospective RCT in parallel groups	401 children with cold or influenza	Nasal saline irrigation delivered via jet flow or fine spray, or added to standard medication vs. standard medication alone. Applied 6 times daily in acute phase and 3 times daily for 12 weeks thereafter.	At the second visit, only 8.7% patients recorded nasal wash complaints, and at the final visit, this dropped to 2.4%. The other reported complaints were burning, bitter taste and nose bleeding.	The saline treatment was well tolerated. Most complaints appeared in the medium jet group and were associated with the stronger flow of the wash.
Süslü, 2009 [28]	Prospective RCT	45 adult subjects after septoplasty	2.3% buffered hypertonic seawater, buffered isotonic saline, unbuffered isotonic saline; irrigation six times daily for 20 days.	No dropouts; no adverse events mentioned.	Buffered isotonic saline group had worse nasal burning VAS score when compared with both buffered hypertonic and nonbuffered isotonic saline solutions.
Ural, 2009 [63]	Observational	132 adult subjects	Control, allergic rhinitis, acute sinusitis and chronic sinusitis groups received two daily doses of hypertonic (3%) or isotonic nasal irrigation for 10 days.	No patients lost to follow up, and no serious side effects or intolerance necessitating cessation of irrigation reported.	Nasal irrigation with isotonic or hypertonic saline can improve mucociliary clearance time in various nasal pathologies.
Gelardi, 2009 [64]	Randomized pilot study	20 adult subjects with acute rhinosinusitis	A nasal syringe (10 mL saline solution, 3 times daily for 14 days) or the Lavonase system (250 mL saline solution sac, twice daily for 14 days).	No adverse effects mentioned.	Nasal irrigation with the Lavonase system was found to be more effective in reducing symptoms and decreasing nasal resistances.
Li, 2009 [65]	RCT	26 children with allergic rhinitis	Saline irrigation, steroid therapy, saline + steroid therapy groups; twice a day for 8 weeks.	No subjects lost to follow up; no adverse events in saline group.	As adjunctive treatment, nasal saline irrigation alleviates the symptoms and signs of allergic rhinitis in children, and decreases use of topical steroids.
Rabago, 2009 [66]	Electronic questionnaire	330 practicing family physicians in Wisconsin, US	Saline nasal irrigation for upper respiratory conditions.	Respondents were not queried directly about perceived safety profile of the treatment.	Analysis showed that 86.7% of respondents have used the treatment as adjunctive care for conditions including chronic rhinosinusitis (91%), acute bacterial rhinosinusitis (67%), seasonal allergic rhinitis (66%), viral upper respiratory infection (59%), other allergic rhinitis (48%), irritant based congestion (48%) and rhinitis of pregnancy (17%).
Cingi, 2010 [16]	Prospective	100 adult subjects with allergic rhinitis	Seawater gel nasal spray in 4-h intervals, two sprays per nostril, from morning till evening for 10 days.	Gel was well-tolerated with no side-effects occurring.	Clinical findings evaluation revealed a statistically significantly decreased rate of nasal congestion and discharge after a 10-day regimen of seawater nasal spray.
Culig, 2010 [67]	RCT	60 patients with chronic rhinosinusitis	Isotonic vs. hypertonic seawater spray solution, applied 3–6 times daily.	No adverse events were observed.	Hypertonic solution was statistically significantly superior to the isotonic for all symptoms.
Miraglia Del Giudice, 2011 [68]	RCT	40 children with seasonal allergic rhinitis	Nasal lavage with Ischia thermal water vs. isotonic saline.	No significant side effects in either group.	Nasal lavage hyper-mineral chloride-sodium water was effective in children with seasonal allergic rhinitis.
Miraglia Del Giudice, 2012 [69]	RCT	34 infants with bronchiolitis	Nebulized normal saline or 3% hypertonic solution in addition to epinephrine and to conventional treatment	Both treatments have an excellent safety profile.	Administration of 3% hypertonic saline is more effective than normal saline in combination with epinephrine in hospitalized children with bronchiolitis.
Hermelingmeier, 2012 [70]	Systematic review and meta-analysis	400 subjects of which 86 were children/adolescent and 45 were pregnant	Different treatments.	No adverse events mentioned, however not all studies included safety outcomes.	Saline nasal irrigation using isotonic solution can be recommended as complementary therapy in allergic rhinitis.
Satdhabudha, 2012 [71]	Prospective RCT	81 children with allergic rhinitis	Buffered hypertonic (1.25%) saline or isotonic saline; nasal irrigation 2 times daily for 4 weeks.	One subject in each study group experienced nasal burning during the first use.	Satisfaction with nasal irrigation was comparable between groups.
Tantilipikorn, 2012 [72]	Prospective RCT	50 adult subjects with chronic rhinosinusitis after endoscopic surgery	Dexpanthenol (Mar Plus) vs. isotonic saline nasal sprays; 4 applications weekly on 1st, 2nd, 4th and 6th postoperative weeks.	Dropout rate was comparable between groups. Three subjects in nasal saline group reported burning sensation.	Product containing seawater (Mar Plus) had better efficacy and comparable safety to nasal saline.
Kumar, 2013 [73]	RCT	50 subjects with chronic sinusitis	Hypertonic (3.5%) or isotonic nasal saline; 10 drops, three times a day in both nostrils for 4 weeks.	None of the patients’ groups reported severe burning sensation. Mild burning sensation was reported by 14.3% in isotonic group and by 57.1% in hypertonic group. Moderate burning sensation was reported by 19% of patients in hypertonic group.	Hypertonic saline nasal solution was more efficacious and well tolerated, and it improved quality of life in patients.
Chen, 2014 [74]	Parallel design with 3 groups	61 children with allergic rhinitis	Nasal irrigation, intranasal corticosteroid, and combined treatment.	No adverse events reported by subjects.	Nasal irrigation and decreased nasal corticosteroids in combination effected a significant improvement in symptoms and signs, and a significant decrease in the mean eosinophile count in nasal secretions were observed at week 12.
Low, 2014 [75]	RCT	74 adult subjects after endoscopic sinus surgery	Normal saline, Ringer’s solution and hypertonic saline group.	No adverse events mentioned.	All groups showed an improvement with treatment in SNOT-20 scores and VAS scores, as well as endoscopic evaluation of mucosa appearance over time, but no improvement of MCC.
Marchisio, 2014 [76]	Questionnaire sent by e-mail	860 primary care paediatricians	Nasal saline irrigation in preschool children.	98.3% of the participating physicians evaluated the treatment as effective and safe.	About 40% of physicians expressed doubts about parental compliance, mainly because of a certain difficulty in administration or the supposed invasiveness of the procedure.
Nguyen, 2014 [77]	Prospective, unblinded, single-arm pilot study	40 subjects with allergic rhinitis	Large-volume low-pressure saline irrigation twice daily for 8 weeks to the ongoing regiment of nasal corticosteroids.	No adverse events reported.	Saline treatment significantly improved QOL, with no significant changes in nasal flows, pattern use of nasal steroids or adverse events.
Pham, 2014 [78]	Retrospective cohort study and cross-sectional survey	144 children with paediatric chronic rhinosinusitis	6 weeks of once-daily nasal irrigation.	The results of a long-term (median of 48 months) follow-up in 54 participants show treatment as safe and well-tolerated.	Nasal irrigation is effective as a first-line treatment for paediatric chronic rhinosinusitis and subsequent nasal symptoms, and reduces need for FESS and CT imaging.
Stoelzel, 2014 [79]	RCT	20 adult subjects with allergic rhinitis	Nasya/Prevalin (a thixotropic nasal gel) vs. isotonic seawater nasal spray; 2 sprays (2 × 0.14 mL) into each nostril.	Three mild AEs were documented in two subjects in the Nasya/Prevalin group (swallowing difficulties, nasal airways obstruction and headache); none related to the application of the investigational product.	There was no difference between the two treatment groups regarding the global assessment of tolerability provided by the investigators or by the subjects.
Wang, 2014 [80]	Prospective, placebo-controlled RCT	60 atopic children with acute sinusitis	Standard treatment (including systemic antibiotics, mucolytics and nasal decongestants) with nasal irrigation with normal saline vs. standard treatment alone.	No significant side effects were recorded in the isotonic saline irrigation group.	There were significant improvements in mean PRQLQ and nPEFR values for the irrigation compared to the non-irrigation group. There was no significant difference in radiographic findings between the groups. The irrigation group recorded significant improvements in eye congestion, rhinorrhea, nasal itching, sneezing and cough symptoms.
Alvarez-Puebla, 2015 [81]	CT	35 adults with asthma	Hypertonic saline (5%, administered by nebulizer) or mannitol.	Treatments were well tolerated.	Mannitol and hypertonic saline behaved similarly at sputum induction.
Koksal, 2016 [82]	Prospective, randomized double-blind trial	109 children under 2 years of age with acute upper respiratory infection	Saline nasal drops (0.9%), seawater nasal drops (2.3%) and control group (no treatment).	No adverse events mentioned.	No significant difference between saline and seawater groups in terms of nasal congestion, but a significant difference between the control group and these two groups.
Bennett, 2015 [18]	RCT, open label, cross-over	12 healthy adults	Hypertonic saline; 2.8% NaCl, 4 mL.	No adverse events mentioned.	Inhaled 2.8% hypertonic saline in normal subjects was associated with a short-lived acceleration of MC, predominately in the central airways.
Bonnomet, 2016 [17]	Randomized, controlled, blinded, in vitro study	Airway epithelial cells obtained from 13 nasal polyp explants	Response (ciliary beat frequency and epithelial wound repair speed) of cells to 3 isotonic nasal irrigation solutions: normal saline 0.9%; non-diluted seawater; and 30% diluted seawater	In vitro study.	Non-diluted seawater obtains the best results on ciliary beat frequency and wound-repair speed vs. normal saline showing a deleterious effect on epithelial cell function.
Grasso, 2018 [83]	Prospective, controlled clinical trial	60 patients with allergic rhinitis	Daily, 5-month treatment with isotonic seawater nasal spray enriched with manganese (4 puffs/day).	No adverse events mentioned.	The treatment significantly decreased the number of episodes of acute allergic rhinitis and increased QOL without the adverse effects of the standard care therapy.
Bergmann, 2019 [84]	Uncontrolled, prospective, longitudinal CT	136 patients with disorders of nose and paranasal sinuses including 11 pregnant women and one nursing mother	Seawater nasal spray (2.7%).	One adverse event reported (epistaxis).	Over the study period (mean 44 days) statistically significant reductions in 10 out of 12 symptoms was found. Only for parameters “impairment of taste” and “impairment of food intake” was no significant change in symptoms observed.
Bogomil’skij, 2019 [85]	Uncontrolled, prospective, longitudinal CT	Children aged 2–5 years with acute infectious rhinitis (some with viral comorbidity)	Aqua Maris spray.	None reported.	Rapid regression of symptoms, such as nasal congestion and snoring, a decrease in the amount of nasal discharge by the 3rd day from the start of drug use and normalization of the rhinoscopic findings by the 5th–7th day of treatment.
Stobbelaar, 2019 [86]	Retrospective study	104 children up to 2 years of age with bronchiolitis in intensive care unit	Nebulised hypertonic saline.	No adverse events mentioned.	In respiratory syncytial virus positive patients, the use of nebulised hypertonic saline was correlated with a decrease in the duration of respiratory support and the length of stay by factors 0.72 and 0.81, respectively.
Craig, 2019 [87]	Prospective, randomised, controlled, double-blind, superiority trial	107 children aged 6 months to 5 years planned to have a nasogastric tube inserted in emergency department	Lidocaine and phenylephrine nasal spray or 0.9% sodium chloride placebo nasal spray, before nasogastric insertion.	Adverse effects occurred in 28% of those who received lidocaine and phenylephrine, and 42% of those who received placebo.	Lidocaine and phenylephrine nasal spray does not reduce procedure-related distress associated with nasogastric tube insertion in young children compared with saline.
Perić, 2019 [88]	Prospective, randomized study	30 patients with aspirin-induced chronic rhinosinusitis undergoing endoscopic sinus surgery	Hypertonic (2.3% NaCl) seawater and isotonic 0.9% NaCl.	Nasal discomforts were detected in two patients in hypertonic seawater group and in two patients in the isotonic group.	Significantly lower total symptom score during the 7th, 14th, 21st and 28th days, lower total endoscopic score on the 21st and 28th days, lower nasal obstruction, facial pain/pressure, headache and trouble sleeping, and lower nasal mucosal oedema, nasal secretion and nasal crusting in patients treated by hypertonic seawater.
Ramalingam, 2020 [89]	Post-hoc secondary analysis of data from the Edinburgh and Lothians Viral Intervention Study	66 adults with upper respiratory tract infection	The intervention group used hypertonic saline at home and performed nasal irrigation and gargling up to 12 times/day. Control arm participants did not use a specific treatment.	None mentioned.	The duration of illness was shorter in the intervention arm in the subset of patients infected with coronavirus (mean 5.6 vs. 8.1 days). The difference in the duration of blocked nose was −3.1 days, cough −3.3 days and hoarseness of voice −2.9 days in favour of hypertonic saline treatment.
Huang, 2021 [90]	In vitro	A 3D reconstituted human nasal epithelium model; mixture of human nasal cells isolated from 14 donors.	Seawater preparation (Stérimar Nasal Hygiene), tissue integrity via transepithelial electrical resistance was measured.	In vitro study.	Treatment did not compromise the integrity of the nasal epithelium in vitro but was effective for removal of foreign particles through MCC increase and for enhancing wound repair on nasal mucosa.
Jiang, 2021 [91]	Multicentre retrospective cohort trial	144 adult subjects with upper respiratory tract infections	Non-drug supportive treatment vs. supportive treatment and nasal irrigation with sea-salt-derived physiological saline.	No adverse events reported.	Seawater group was statistically significantly superior in terms of nasal congestion, nasal discharge, sleep quality and appetite, but not for cough and fatigue.

**Table 4 marinedrugs-20-00330-t004:** Overview of review articles with saline and/or saltwater.

Study	Design	Indication(s)	Intervention(s)	Remarks
Papsin, 2003 [27]	Literature review	Rhinosinusitis, allergic rhinitis, postoperative irrigation, common cold	Nasal irrigation as an adjunct treatment	The procedure has been used safely by both adults and children and has no documented serious adverse effects. Trials indicate that patients treated with nasal irrigation rely less on other medications, and that some postsurgical patients tend to require fewer visits to physicians. Both effects are likely to have desirable economic consequences for patients and the health care system.
Brown, 2004 [93]	Literature review	(Chronic) sinusitis, sinonasal conditions, rhinitis, postoperative patients	Isotonic and hypertonic saline, buffered/unbuffered solutions, additives such as antibacterial or antifungal agents, home recipes vs. manufactured solutions	Nasal irrigations are an important component in the management of most sinonasal conditions. Authors note a disparity of opinion about the effects of irrigations on ciliary beat frequency and mucociliary clearance and controversy concerning irrigation tonicity and the use of additives to the irrigating solution.
Harvey, 2007 [94]	Review (Cochrane)	Chronic sinusitis	Randomised controlled trials in which saline was evaluated in comparison with either no treatment, a placebo, as an adjunct to other treatments or against treatments. The comparison of hypertonic versus isotonic solutions.	Saline irrigations are well tolerated. Although minor side effects are common, the beneficial effect of saline appears to outweigh these drawbacks for the majority of patients. The use of topical saline could be included as a treatment adjunct for the symptoms of chronic rhinosinusitis.
Kassel, 2010 [95]	Review (Cochrane)	Upper respiratory tract infections	RCTs comparing topical nasal saline treatment to other interventions in adults and children with clinically diagnosed acute URTIs.	Three RCTs (618 participants) were included. Most results showed no difference between nasal saline treatment and control. However, there was limited evidence of benefit with nasal saline irrigation in adults. Minor discomfort was not uncommon and 40% of babies did not tolerate nasal saline drops.
Zhang, 2008 [96]	Review (Cochrane)	Acute bronchiolitis in infants	Nebulized hypertonic saline alone or in conjunction with bronchodilators as an active intervention in infants with acute bronchiolitis.	Current evidence suggests nebulized 3% saline may significantly reduce the length of hospital stay among infants hospitalized with non-severe acute viral bronchiolitis and improve the clinical severity score in both outpatient and inpatient populations.
Adappa, 2012 [97]	Literature review	Rhinosinusitis	Saline irrigation (hypertonic vs. physiologic), saline spray, antibiotics, topical steroids, topical antifungal treatment, anti IL-5 treatment	Physiologic saline irrigation is beneficial in the treatment of symptoms of CRS. Low-level evidence supports the effectiveness of topical antibiotics in the treatment of CRS. The use of topical antifungals is not supported by the majority of studies. Intranasal steroids are beneficial in the treatment of CRS with nasal polyposis. There is insufficient evidence to demonstrate a clear overall benefit for topical steroids in CRS without nasal polyposis.
Chirico, 2014 [98]	Literature review	Nasal congestion in infants and children	Nasal saline	The use of isotonic and hypertonic saline solutions is a valuable non-pharmacological treatment for nasal congestion in children, especially by improving mucociliary clearance and reducing the use of medications (antihistamines, decongestants, antibiotics, corticosteroids) during the treatment of URTIs. They are well tolerated and can be recommended for infants.
Bastier, 2015 [20]	Overview of randomized clinical trials	Different sinonasal pathologies and postoperative care	Different treatments compared to nasal irrigation including rhinocorticoids, antihistamines, buffered, unbuffered, alkaline hypertonic and isotonic saline	Large-volume low-pressure nasal irrigation using undiluted seawater seems, according to the present state of knowledge, to be the most effective protocol.
Chong, 2016 [99]	Review (Cochrane)	Chronic rhinosinusitis	Studies with follow-up periods of at least three months comparing saline delivered to the nose by any means (douche, irrigation, drops, spray or nebuliser) with placebo, no treatment or other pharmacological interventions	The evidence suggests that there is no benefit from a low-volume nebulised saline spray as opposed to intranasal steroids. There is some benefit from daily, large-volume (150 mL) saline irrigation with a hypertonic solution when compared with placebo.
Baron, 2016 [39]	Literature review	Bronchiolitis in infants	Hypertonic saline	Authors agree with the AAP guidelines regarding the use of nebulised hypertonic saline to reduce bronchiolitis scores and length of stay for infants with bronchiolitis who are expected to be hospitalised for more than 72 h.
Madison, 2016 [100]	Literature review	Allergic rhinitis in children	Nasal saline irrigation vs. intranasal corticosteroids	Intranasal steroids are more effective than nasal saline alone in reducing symptoms of allergic rhinitis in children. However, combination therapy further improves symptom reduction.
Kanjanawasee, 2018 [101]	Systematic search with Ovid MEDLINE, Scopus, PubMed and Google Scholar	Sinonasal diseases, including rhinitis and rhinosinusitis	Hypertonic vs. isotonic saline	Nine studies (740 patients) were included. Hypertonic nasal irrigation brought greater benefits than isotonic treatment in symptom reduction; however, no difference was shown in SNOT-20 improvement. Effects favouring hypertonic solution were greater in patients with rhinitis compared with rhinosinusitis; in patients under the age of 18 years; in saline irrigation using high volume compared with low volume and in saline irrigation with hypertonicity of <3% and hypertonicity of 3–5% compared with hypertonicity of >5%. No major adverse effects were reported.
Li, 2019 [102]	Systematic review and meta-analysis literature following the PRISMA guidelines	Allergic rhinitis in children	Hypertonic saline nasal irrigation	Hypertonic saline treatment improved patients’ nasal symptom scores and significantly lowered rescue antihistamine use rate. Analyses comparing hypertonic with isotonic saline nasal irrigation found better nasal symptom scores in hypertonic group, although the antihistamine use and adverse-effect rates were similar between groups.
King, 2019 [103]	Literature review with evidence for each of the indications	Chronic sinusitis, allergic rhinitis, acute URTI	Saline solutions, dependent on the indication studied	Saline nasal irrigation is recommended as an adjunct therapy for common colds/rhinosinusitis, chronic sinusitis, allergic rhinitis and after nasal surgery. It appears to be safe and generally well tolerated, even for children. The use of SNI has the potential to reduce the number of antibiotic prescriptions for acute and chronic sinus infections, and improve outcomes for patients.

## Data Availability

Not applicable.

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
