# Peer review of "The Role of Seawater and Saline Solutions in Treatment of Upper Respiratory Conditions"

_marinedrugs, 2022, doi:10.3390/md20050330_

Round 1

Reviewer 1 Report

I appreciate the opportunity to review the manuscript for publication in MDPI Marine Drugs.

There are several points that need additional comments and amendments related to the topics.

It is advisable for the authors to describe sea water compositions in relation to world-wide climate changes in the 21th century.

L109: “Depending on the place of application, this leads to accumulation of liquid in the lumen and increased mucociliary clearance [11,12]. The mechanism has been proven both in vitro [13,14] and in vivo [15,16]. Reduction of swelling (oedema) is seen in submucosal tissue, while the immediate effect of excess liquid in nasal lumen is mechanical cleaning of mucus, crusts and debris. Imminently following is the change of the state of the mucus from gel to sol [17].”

In this paragraph about physiological effects of seawater, the cited references are mainly about lower airways and about specific disease (cystic fibrosis). The authors had better refer more adequate refs specific to nose (nasal allergy) and paranasal sinuses (chronic rhinosinusitis).  

L120: “Mucociliary transit time (MTT; the time needed for a compound to be transported a certain distance within the respiratory system),”

The references cited in this para do not mention beneficial effects of seawater.

L170: “intranasal treatment with saline and sea-water preparations in form of either drops, spray, nebulizer or irrigation is considered to be very safe.”

Again, the authors had better discriminate clearly between physiological saline and sea water.

Table 5. Overview of safety and efficacy conclusions from studies with saline and/or saltwater.

I understand that most studies listed employ medical therapy such as topical decongestants, nasal steroids, or mucolytic agents as well other than topical saline irrigation. They should also be stated.

L273: “In our paper we present a comprehensive body of evidence why sea-water is superior to saline for SNI in general as well as for the wide variety of clinical indications such as infectious diseases of the upper respiratory tract, allergic rhinitis, postoperative care etc.”

I do not think the authors can draw these concrete conclusions based on previous literature. At least, there are no description on EBM levels and recommendation grades in Tables 5 and 6.

Author Response

Dear Reviewer,

We have prepared a revised version of our manuscript according to the reviewers’ comments and accordingly provide below the answers to their comments point by point.

We hope now that our manuscript meets the standards and requirements to be published in Marine Drugs.

Kind regards,

PhD Danijela Štanfel

Jadran Galenski laboratorij d.d.

Svilno 20

51000 Rijeka

Croatia

Author Response

(The authors gave the same response as above.)

Reviewer 3 Report

This review is very interesting and worth in the field of Otolaryngology. If possible, please change Respiratory Conditions to Upper Respiratory Conditions, since in this review, the authors mainly discuss the influence of saline/seawater in the treatment of upper airway diseases.

Author Response

(The authors gave the same response as above.)

Round 2

Reviewer 1 Report

The manuscript has been revised and improved in accordance with the reviewers’ comments.

Author Response

Dear Reviewer,

Thanks a lot for all your comments. We hope now that our manuscript meets the standards and requirements to be published in Marine Drugs.

Kind regards, PhD Danijela Štanfel

Jadran Galenski laboratorij d.d.

Svilno 20

51000 Rijeka

Croatia

Dear Editor,

We have prepared a revised version of our manuscript according to the Academic editor comments and accordingly provide below the answers to their comments point by point.

We hope now that our manuscript meets the standards and requirements to be published in Marine Drugs.

Kind regards, PhD Danijela Štanfel
